# MimicTalk: Mimicking a personalized and expressive 3D talking face in minutes

**Zhenhui Ye** [†‡1,2]   **Tianyun Zhong** [†‡1,2]   **Yi Ren** [2]   **Ziyue Jiang** [‡1,2]   **Jiawei Huang** [‡1,2]
**Rongjie Huang** [1]   **Jinglin liu** [2]   **Jinzheng He** [1]   **Chen Zhang** [2]   **Zehan Wang** [1]
**Xize Chen** [1]   **Xiang Yin** [2]   **Zhou Zhao** [*1]
[1]Zhejiang University,   [2]ByteDance
{zhenhuiye,zhaozhou}@zju.edu.cn,
{ren.yi,yinxiang.stephen}@bytedance.com

## Abstract

Talking face generation (TFG) aims to animate a target identity's face to create realistic talking videos. Personalized TFG is a variant that emphasizes the perceptual identity similarity of the synthesized result (from the perspective of appearance and talking style). While previous works typically solve this problem by learning an individual neural radiance field (NeRF) for each identity to implicitly store its static and dynamic information, we find it inefficient and non-generalized due to the per-identity-per-training framework and the limited training data. To this end, we propose MimicTalk, the first attempt that exploits the rich knowledge from a NeRF-based person-agnostic generic model for improving the efficiency and robustness of personalized TFG. To be specific, (1) we first come up with a person-agnostic 3D TFG model as the base model and propose to adapt it into a specific identity; (2) we propose a static-dynamic-hybrid adaptation pipeline to help the model learn the personalized static appearance and facial dynamic features; (3) To generate the facial motion of the personalized talking style, we propose an in-context stylized audio-to-motion model that mimics the implicit talking style provided in the reference video without information loss by an explicit style representation. The adaptation process to an unseen identity can be performed in 15 minutes, which is 47 times faster than previous person-dependent methods. Experiments show that our MimicTalk surpasses previous baselines regarding video quality, efficiency, and expressiveness. Source code and video samples are available at `https://mimictalk.github.io`.

## 1   Introduction

Audio-driven talking face generation (TFG) (Prajwal et al., 2020; Hong et al., 2022; Tian et al., 2024; Xu et al., 2024) is a cross-modal task that leverages multi-modal knowledge from speech, vision, and computer graphics to animate a target identity's face given arbitrary driving audio, aiming to create lifelike talking videos or interactive avatars. Personalized TFG (Suwajanakorn et al., 2017; Thies et al., 2020a; Guo et al., 2021; Lu et al., 2021b) is a variant of TFG with several real-world applications, such as video conferencing and audio-visual chatbots, in which we emphasize that the generated results should have excellent perceptual similarity to a specific individual (from the perspective of both visual quality and expressiveness). In this setting, a video clip of the target identity (from several seconds to minutes) is provided as a detailed reference for the target person's

---

† Equal contribution.
‡ Interns at ByteDance.
∗ Corresponding author.

38th Conference on Neural Information Processing Systems (NeurIPS 2024).

personalized attributes, such as geometry shape (Mildenhall et al., 2021; Kerbl et al., 2023) and talking style (Wu et al., 2021; Ma et al., 2023; Tan et al., 2023). To meet the quality requirement of high perceptual identity similarity, the community has focused on identity-dependent methods (Tang et al., 2022; Li et al., 2023a; Xu et al., 2023), in which an individual model is trained from scratch for each target person's video. The primary reason for the prevailing of these identity-dependent approaches is that during the training process, the person-specific model can implicitly memorize nuanced personalized details of the target person's video, such as their speaking style and subtle expressions, which are hard to represent explicitly with hand-crafted conditions. As a result, these approaches could train small-scale person-dependent models, achieving high identity similarity and expressive results.

However, the identity-dependent methods (Tang et al., 2022; Li et al., 2023a) face two well-known challenges: (1) the first is weak generalizability. The limited training data scale in the per-identity-per-training restricts the generalizability to out-of-domain (OOD) conditions during inference. For example, lip-sync performance may degrade when driven by OOD audio (from a different language or speaker), and the rendering may fail when synthesizing OOD expressions (such as large yaw). (2) the second is low training and sample efficiency. Since the identity-dependent model needs to be trained from scratch on the target person video and no prior knowledge can be enjoyed, the training process can take more than several hours (low training efficiency), and it often requires over 1-minute length of training data to achieve reasonable lip-sync results (low sample efficiency). By contrast, another line of works, the identity-agnostic (or, say, one-shot) TFG methods (Zhou et al., 2020; Wang et al., 2021; Zhao and Zhang, 2022; Li et al., 2023b), which incorporate various identities' video during training and only use one source image during inference, can achieve good generalizability to OOD conditions since various audio/expressions have been seen during the training. However, due to the limited information in the single input image, the one-shot methods cannot exploit the rich samples of the target identity to imitate its personalized attributes. Therefore, it is promising to bridge the gap between person-agnostic one-shot methods and person-dependent small-scale models to achieve expressive, generalized, and efficient personalized TFG.

Based on the observation above, our key motivation is to dig out the prior knowledge of pre-trained person-agnostic TFG models (Li et al., 2024; Chu et al., 2024; Ye et al., 2024) to better mimic the personalized attributes of target identities. We propose MimicTalk, a high-quality and expressive personalized TFG framework, which utilizes the generalizability power of a large generic TFG model to improve the efficiency and robustness of TFG and devises a talking style-controllable motion generator to obtain results of high expressiveness and personalized characteristics. Specifically, (1) our method starts with a 3D person-agnostic generic model. This is due to the finding that the widely used warping-based methods cannot well handle large movements, and state-of-the-art person-dependent methods have turned to Neural Radiance Field (NeRF) or Gaussian Splatting, which are advanced neural rendering techniques with strong 3D prior, to achieve video consistency and geometric realism. Inspired by this, we resort to a recent NeRF-based one-shot TFG method to provide a 3D-aware base model in our framework. (2) Given a pre-trained one-shot model, we design a static-dynamic (SD)-hybrid adaptation pipeline for an efficient and stable finetuning process. Specifically, we propose a tri-plane inversion method to learn a personalized 3D face representation to store the static texture details; we also first introduce low-rank adaptation (LoRA) in TFG to adapt the dynamic characteristics of the target speaker, which are difficult to model explicitly. (3) We note that generating motion of personalized talking styles is vital for perceptual reality in audio-driven TFG. Although finetuning the motion generation module on the target individual dataset can allow the network to implicitly memorize the target talking style, it leads to high computation costs and training instability. To address this, we propose an in-context-style audio-to-motion (ICS-A2M) model capable of mimicking the in-context zero-shot talking style. The ICS-A2M model is facilitated by a newly proposed training paradigm named audio-guided motion-infilling, which encourages the model to denoise the partially masked motion track by exploiting the implicit talking style revealed in the unmasked motion track. We also incorporate conditional flow-matching into the co-speech motion generation task, achieving state-of-the-art lip-sync and style-mimicking accuracy.

The contributions of the paper are summarized as follows:

- We are the first work that considers utilizing 3D person-agnostic models for personalized TFG. We propose an SD-hybrid pipeline for efficient and high-quality adaptation to learning the static and dynamic characteristics of the target speakers.

- We propose the ICS-A2M model, which achieves high lip-sync quality and in-context talking style mimicking audio-driven TFG.

- Our MimicTalk only requires a few seconds long reference video as the training data and several minutes for training. Experiments show that our MimicTalk surpasses previous person-dependent baselines in terms of both expressiveness and video quality while achieving 47x times faster convergence.

## 2 Related Work

Our work is mainly about personalized and expressive talking face generation. We discuss the related works in the field of talking face generation and expressive co-speech facial motion generation, respectively.

### 2.1 Talking Face Generation

Based on different real-world applications, talking face generation (TFG) methods have been majorly divided into two settings: identity-agnostic and identity-dependent. The identity-agnostic methods focus on the one-shot scenario: the model is provided with only one image and aims to animate it to create a video. These approaches hence train a large-scale generic model with a large amount of identity video data so that it can generalize to unseen photos during the inference stage. On the other hand, identity-dependent methods focus on achieving better video quality for a specific speaker: typically using a segment of the target user's video as training data and expecting the model to mimic the personalized features of that speaker, known as personalized TFG. These two lines of work have evolved independently over the years and have developed significantly different methodologies. (1) As for the **person-agnostic** methods, the earliest works (Chung et al., 2017; Prajwal et al., 2020; Wang et al., 2022) typically adopt a pixel-to-pixel framework (Isola et al., 2017) or generative adversarial network (GAN) setting to generate the result, which results in training instability and bad visual quality. Then, the majority of prior works (Averbuch-Elor et al., 2017; Ren et al., 2021; Wang et al., 2021; Hong et al., 2022; Zhao and Zhang, 2022; Hong and Xu, 2023; Liang et al., 2024; Jiang et al., 2024b) resort to a dense warping field (Siarohin et al., 2019) to deform the pixels of the source image given the 3D-aware keypoints extracted from the driving resource. The warping-based method achieves better image fidelity, yet due to a lack of 3D prior knowledge, it occasionally produces warping and distortion artifacts. Recently, to handle these artifacts, some work (Zeng et al., 2022; Sun et al., 2023; Li et al., 2023b,c; Ye et al., 2024) propose one-shot NeRF-based methods by learning to reconstruct a 3D face representation (tri-plane) (Chan et al., 2022) from the source image. (2) As for the **person-dependent setting** (Thies et al., 2020b; Yi et al., 2020; Lu et al., 2021a; Ye et al., 2022), which trains an individual model on a specific video of the target identity, the recent works (Guo et al., 2021; Yao et al., 2022; Tang et al., 2022; Ye et al., 2023; Li et al., 2023a) are mainly based on NeRF for its high image fidelity and realistic 3D modeling. Despite the NeRF-based person-specific method achieving the best video quality and identity similarity among all TFG methods, it typically requires hours of training, and its generalizability to driving conditions is hampered by limited training data. To our knowledge, Mimictalk is the first work that considers bridging the gap between NeRF-based person-agnostic and person-dependent methods to improve the task of personalized TFG.

### 2.2 Expressive Facial Motion Generation

The recent advance in neural rendering has significantly improved the image quality, temporal consistency, and stability of the synthesized video, making expressiveness the next focus of the TFG community. The critical challenge in achieving high expressiveness is to generate high-quality facial motion from audio content. This requires the generated motion sequence to not only synchronize with the audio track but also reflect a consistent and expressive talking style similar to the target speaker. Some studies implicitly model this process in a deterministic and end-to-end audio-to-image model (Prajwal et al., 2020; Guo et al., 2021). For better controllability and audio-lip synchronization, other works propose explicitly modeling the audio-to-motion mapping using external generative models (Thies et al., 2020b; Ye et al., 2023). However, these studies do not explicitly model the speaker's talking style. To address this, Wu et al. (2021) and Ma et al. (2023) developed a style vector extracted from arbitrary motion sequences to achieve explicit talking style control.

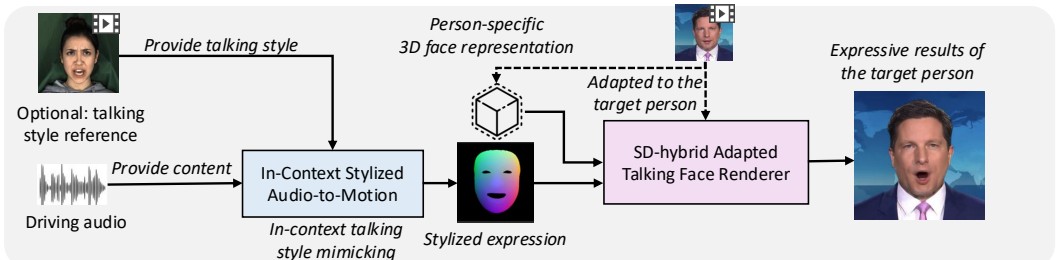

Figure 1: The inference process of MimicTalk. We use an in-context stylized audio-to-motion model to produce expressive facial motion mimicking the talking style of a reference video. Then, a personalized renderer could render high-quality talking face videos that mimic the static and dynamic visual attributes of the target identity.

EMMN (Tan et al., 2023) disentangles expression styles and lip motion, constructing a memory bank to produce lip-synced videos with vivid expressions. A concurrent work, VASA-1 (Xu et al., 2024), proposes to add previous audio/motion latent as the input of the latent diffusion model to keep temporal consistency. We can see that most previous methods rely on intermediates to represent the talking style Jiang et al. (2024a), risking information loss. In contrast, our method first achieves in-context-learning (ICL) talking style control, better preserving the target identity's talking style.

## 3  MimicTalk

As shown in Fig. 1, MimicTalk is a personalized and expressive 3D talking face generation framework, in which the personalized renderer inherits the rich facial knowledge from a person-agnostic generic model (discussed in Sec. 3.1) via a static-dynamic (SD)-hybrid adaptation pipeline (in Sec. 3.2). In Sec. 3.3, we propose an in-context stylized audio-to-motion (ICS-A2M) model to generate personalized facial motion, which is necessary to achieve high expressiveness in the generated video. We describe the design and training process in the following sections. Due to space limitations, we provide technical details in Appendix B.

### 3.1  NeRF-based Person-Agnostic Renderer

The previous personalized TFG works trained a separate model on the target speaker to memorize personalized information of the target person, in which NeRF is typically used as the underlying technology to better store the target speaker's geometry, texture, and other information. We aim to leverage the prior knowledge of pre-trained person-agnostic TFG models to achieve greater generalizability and efficiency than previous person-dependent methods. To this end, we resort to recent one-shot NeRF-based TFG works (Li et al., 2023b, 2024; Ye et al., 2024; Chu et al., 2024) to construct a person-agnostic TFG model. Specifically, We build our base model on the basis of Real3D-Portrait (Ye et al., 2024) with its official implementation [1]. As shown in Fig. 2, the first step is reconstructing a canonical 3D face (formulated as tri-plane representation (Chan et al., 2022)) from the target person's source image $\mathbf{I}_{\text{src}}$:

$$\mathbf{P}_{\text{cano}} = \texttt{FaceRecon}(\mathbf{I}_{\text{src}}), \tag{1}$$

where $\texttt{FaceRecon}$ is an image-to-3D face reconstruction model based on SegFormer (Xie et al., 2021) that transforms the input image into a tri-plane representation Chan et al. (2022). Then a light-weight SegFormer-based motion adapter could control the facial expression in the 3D face, and a volume renderer of Mip-NeRF (Barron et al., 2021) could render the dynamic talking face of arbitrary head pose by controlling the camera:

$$\mathbf{I}_{\text{raw}} = \texttt{VolumeRenderer}(\mathbf{P}_{\text{cano}} + \texttt{MotionAdapter}([\mathbf{PNCC}_{\text{src}}, \mathbf{PNCC}_{\text{tgt}}]), \mathbf{cam}_{\text{tgt}}), \tag{2}$$

where $\mathbf{PNCC}_{\text{src}}$ and $\mathbf{PNCC}_{\text{tgt}}$ are projected normalized coordinate codes (Li et al., 2023b), which is rasterized from the 3DMM face (Paysan et al., 2009) of the source and target image, and $\mathbf{cam}_{\text{drv}}$

---

[1] https://github.com/yerfor/Real3DPortrait

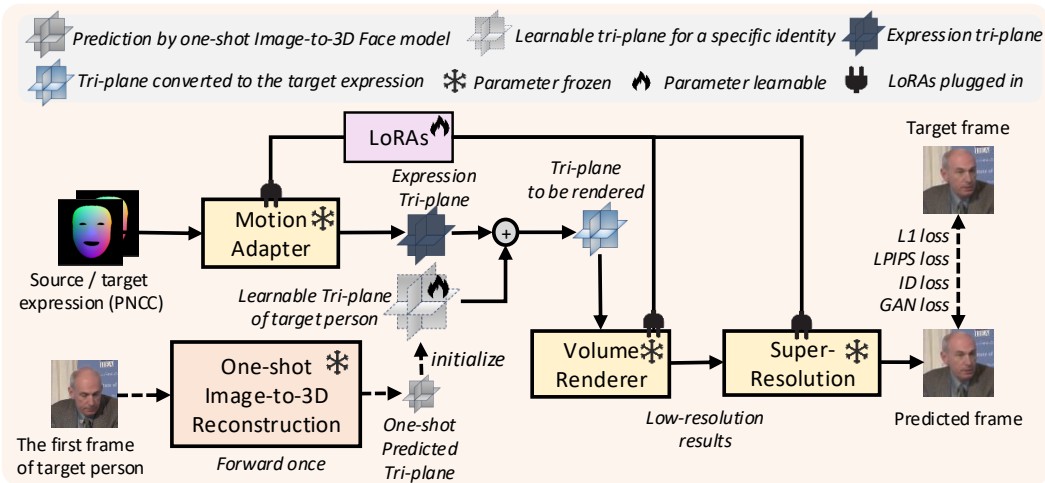

Figure 2: The training process of the personalized TFG renderer via the static-dynamic (SD)-hybrid adaptation pipeline. We adopt a pretrained one-shot person-agnostic 3D TFG model as the backbone, then fine-tune a person-dependent 3D face representation to memorize the *static* geometry and texture details. We also inject LoRA units into the backbone to learn the personalized *dynamic* features.

is the driving camera to control the head pose. $\mathbf{I}_{raw}$ is the volume-rendered low-resolution image, which is further processed by a super-resolution module to generate the final high-resolution result:

$$\mathbf{I}_{pred} = \texttt{SuperResolution}(\mathbf{I}_{raw}) \tag{3}$$

We provide detailed network structure of the `FaceRecon`, `MotionAdapter`, `VolumeRenderer`, and `SuperResolution` of person-agnostic renderer in Fig. 5 of Appendix B. For more details about the one-shot person-agnostic base model, please refer to (Ye et al., 2024).

## 3.2 Static-Dynamic-Hybrid Identity Adaptation

With the pre-trained person-agnostic renderer introduced in Sec. 3.1, we could synthesize talking face videos for unseen identities without any adaptation. However, there exists a significant identity similarity gap between the un-tuned renderer and previous person-dependent methods, which is reflected in two aspects: (1) The *static similarity*, which measures whether the generated frame has same texture (e.g., wrinkles, teeth, and hair) or geometry details as the target identity; (2) The *dynamic similarity*, which describes the relationship between the input motion condition and the facial muscle/torso movement in the output facial image. To be more intuitive, trained on the large-scale talking face dataset with various speakers, our person-agnostic model learns a statistically averaged motion-to-image mapping for face animation, which produces talking person videos that are semantically correct yet lack personalized characteristics. By contrast, the previous person-dependent methods, by overfitting the model on one target identity, inherently learn the personalized motion-to-image mapping in the model. Based on the observation above, we propose an efficient *static-dynamic-hybrid (SD-Hybrid) adaptation pipeline* to achieve good static/dynamic identity similarity, which is shown in Fig. 2.

**Tri-Plane Inversion for Static Similarity.** The canonical 3D face representation $\mathbf{P}_{cano}$, which is extracted from the source image by the pre-trained 3D face reconstruction model (Ye et al., 2024), stores all static properties of the target identity (i.e., geometry and texture information). We find that the information loss in this feed-forward image-to-3D transform is the primary reason for the inferior static similarity in our method. To this end, inspired by previous GAN-inversion methods (Roich et al., 2021), we propose a tri-plane inversion technique, which regards the canonical 3D face representation as a learnable parameter and optimizes it to maximize the static identity similarity. To be specific, as shown in Fig. 2, when adapting our model to a specific identity given a video clip, we initialize the learnable tri-plane with the first frame's image-to-3D prediction, then optimize the tri-plane alongside other parameters.

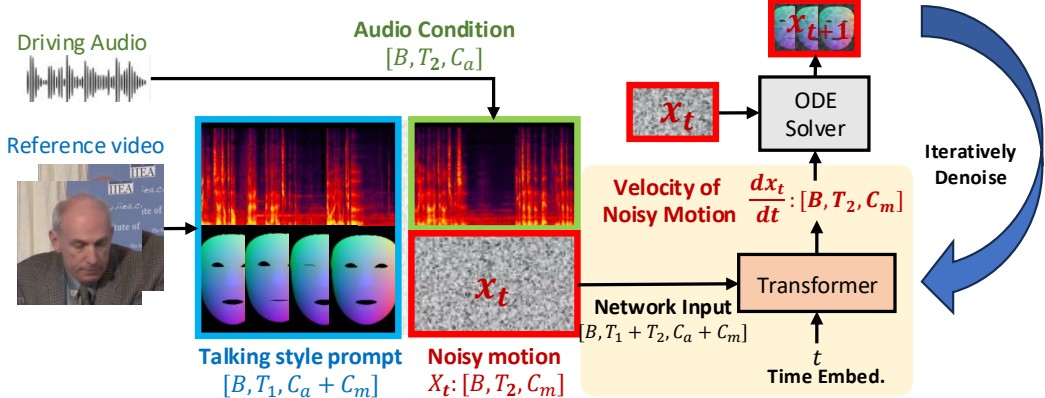

Figure 3: The process of in-context stylized motion prediction. For the training process please refer to Fig. 7.

**Injecting LoRAs for Dynamic Similarity.** As for dynamic similarity, since the person-agnostic model learns averaged motion-to-image mapping in the multi-speaker talking face dataset, we need to adapt the generic model specifically to the target person's video to learn its personalized facial dynamics. A naive solution is to directly fine-tune the whole model or the last few layers on the target person's video. However, considering the large model capacity and the small data scale of the target person's video, it faces several challenges, such as high GPU memory footprints, training instability, and catastrophic forgetting. To this end, we resort to low-rank adaptation (LoRA) (Hu et al., 2021), which was initially proposed for efficient adaptation of language models and recently extended to computer vision applications such as text-to-image synthesis (Rombach et al., 2021). As shown in Fig. 6 of Appendix B.2, LoRAs can be conveniently plugged into our person-agnostic model by injecting a low-rank learnable matrix to every linear layer and convolution kernel. All pre-trained parameters in the person-agnostic model are fixed, and only the LoRAs are updated during training.

**Adaptation Process.** As shown in Fig. 2, with the SD-hybrid design, the rendering process of the predicted image can be expressed as:

$$\mathbf{I}_{\text{pred}} = \text{SR}(\text{VolumeRenderer}(\mathbf{P}^*_{\text{cano}} + \text{MotionAdapter}([\mathbf{PNNC}_{\text{src}}, \mathbf{PNCC}_{\text{tgt}}]; \theta^*_1), \mathbf{cam}_{\text{drv}}; \theta^*_2); \theta^3_3),$$
(4)

where $\mathbf{P}^*_{\text{cano}}$ is the learnable 3D face representation, which is initialized from the first-frame prediction by the pre-trained 3D face reconstruction model. SR denotes the super-resolution module, $\theta^*_1, \theta^*_2, \theta^*_3$ are the learnable LoRA parameters injected in the volume renderer, motion adapter, and the super-resolution module, respectively.

During adaptation, a short video clip of a specific identity is utilized as the training data, and only the personalized tri-plane and LoRAs are updated. The training loss of the SD-hybrid adaptation is:

$$\mathcal{L}_{\text{SD-Hybrid}} = \mathcal{L}_1 + \lambda_{\text{LPIPS}} \cdot \mathcal{L}_{\text{LPIPS}} + \lambda_{\text{ID}} \cdot \mathcal{L}_{\text{ID}},$$
(5)

where $\mathcal{L}_1, \mathcal{L}_{\text{LPIPS}}, \mathcal{L}_{\text{ID}}$, denotes L1 loss, LPIPS loss by VGG16 (Simonyan and Zisserman, 2014), and identity loss by VGGFace (Cao et al., 2018), respectively. We set the learning rate to 0.001, $\lambda_{\text{LPIPS}} = 0.2, \lambda_{\text{ID}} = 0.1$. Thanks to the static-dynamic-hybrid design, our method achieves good identity similarity while enjoying a fast and low-memory-cost adaptation process compared to existing person-dependent methods, illustrated in Table 1. Besides, we demonstrate that our SD-Hybrid pipeline achieves good training and sample efficiency in Fig. 4.

### 3.3 In-Context Stylized Audio-to-Motion

In the above sections, we have proposed a unified framework for motion-conditioned talking face generation. Then, we present **in-context stylized audio-to-motion (ICS-A2M)** model to generate personalized facial motion for audio-driven scenarios.

**Audio-guided Motion Filling Task**   Inspired by the success of in-context learning methods in large language models (Liu et al., 2023) and text-to-speech synthesis (Le et al., 2023), we devise an audio-guided motion infilling task. We visualize the audio-guided motion-infilling task's detailed training and inference pipeline in Fig. 7 of Appendix B.3. Specifically, the audio-motion pairs are temporally aligned and channel-wise concatenated and processed by the audio-to-motion model. During training, we randomly masked several segments in the motion track and trained the model with the motion reconstruction error on the masked segments. Since the model is provided with surrounding unmasked motion and the complete audio track, it learns to exploit the talking style in the motion context to predict the masked co-speech motion more accurately. During inference, as shown in Fig. 3 (a), we could concatenate the reference audio-motion pair as the talking style prompt, arbitrary driving audio as the condition, and a noisy placeholder for predicted motion as the input of the model. This way, the model could predict the audio-synchronized facial motion with the talking style provided in the style prompt.

**Audio-to-Motion Flow Matching Model**   We first introduce an advanced generative model, flow matching (Lipman et al., 2023), to generate expressive facial motions. The model is parameterized by a transformer $\theta$ and predicts a flow field $\phi$ that stores the velocity $\mathbf{v}_t = \frac{d\mathbf{x}_t}{dt}$ of the data points $\mathbf{x}_t$ pointing towards the target distribution. Similar to diffusion models, the time step $t$ is increasing from 0 to 1 during the inference process, i.e., we have prior data points $\mathbf{x}_0 \sim N(0,1)$ and the target data points $\mathbf{x}_1 \sim q(\mathbf{x})$, where $q(\mathbf{x})$ denotes the ground truth data distribution. Due to space limitation, we provide detailed preliminaries of flow matching in Appendix B.3.2. To be intuitive, we visualize the forward process of our flow-matching-based audio-to-motion model in Fig. 3 (a). The network input is the same as what we defined in the previous paragraph (a concatenation of the style prompt, audio condition, and noisy motion $\mathbf{x}_t$). The model's output is the velocity $\mathbf{v}_t$ of the noisy motion $\mathbf{x}_t$. And we could train the model with the conditional flow matching (CFM) objective:

$$\mathcal{L}_{\text{CFM}} = \mathbb{E}_{t,q(x),p(x|x_1)}||\mathbf{u}_t(\mathbf{x}|\mathbf{x}_1) - \mathbf{v}_t(\mathbf{x}, \mathbf{a}, \mathbf{s}; \theta)||_2^2, \tag{6}$$

where $\mathbf{v}_t(\mathbf{x}, \mathbf{a}, \mathbf{s}; \theta)$ is the predicted velocity by flow matching model of parameter $\theta$, $\mathbf{a}$ and $\mathbf{s}$ is th audio condition and style prompt, and $\mathbf{u}_t(\mathbf{x}|\mathbf{x}_1)$ is the ground truth velocity of the optimal transport (OT) path (Le et al., 2023) given the target motion $\mathbf{x}_1$ and the noisy motion $\mathbf{x}_t$, which is exactly the unit vector pointing from $\mathbf{x}_t$ to $\mathbf{x}_1$.

Apart from the flow matching objective, we find it necessary to add a lip-sync loss on the denoised sample $\hat{\mathbf{x}}_1$ to generate accurate and expressive facial motion. Specifically, we first solve the ODE problem in Eq. 9 to obtain the predicted sample $\hat{\mathbf{x}}_1$ using the predicted velocity $\mathbf{v}_t(\mathbf{x}, \mathbf{a}, \mathbf{s}; \theta)$, then feed it into a pre-trained audio-expression SyncNet (Ye et al., 2023) to obtain a discriminative sync loss $\mathcal{L}_{\text{sync}}$ that measure the synchronization between the input audio and the predicted expression. Therefore, the training loss of the ICS-A2M model is:

$$\mathcal{L}_{\text{ICS-A2M}} = \mathcal{L}_{\text{CFM}} + \lambda_{\text{sync}} \cdot \mathcal{L}_{\text{sync}}, \tag{7}$$

where $\lambda_{\text{sync}} = 0.05$. During inference, as visualized in Fig. 3(b), now that the A2M model could predict the velocity field of the data point, we could iteratively push the data from the Gaussian distribution to the target distribution by solving the ordinary differential equation in Eq. 9 of Appendix B.3.2 and finally obtain the predicted motion $\hat{x}_1$ to drive the personalized renderer in Sec. 3.2.

**Classifier-Free Guidance for Enhancing Style Mimicking**   Classifier-free guidance (CFG) is a popular inference method that manipulates the condition strength during the sampling process of conditional diffusion models. While CFG has become a standard technique in text-to-image Rombach et al. (2021) and other weak-conditioned generation tasks, we found it also helps enhance the style mimicking quality of our stylized audio-to-motion model. Specifically, we mix the two network predictions with/without style prompt to construct the CFG velocity $\mathbf{v}_t^{\text{CFG}}$:

$$\mathbf{v}_t^{\text{CFG}} = \mathbf{v}(\mathbf{x}, \mathbf{a}, \mathbf{s}; \theta) + w \cdot (\mathbf{v}(\mathbf{x}, \mathbf{a}, \mathbf{s}; \theta) - \mathbf{v}(\mathbf{x}, \mathbf{a}, \mathbf{0}; \theta)), \tag{8}$$

where $w$ is the CFG scale and we empirically set $w = 2$. $\mathbf{v}(\mathbf{x}, \mathbf{a}, \mathbf{s}$ is the network prediction with both audio condition and style prompt as the input, and $\mathbf{v}(\mathbf{x}, \mathbf{a}, \mathbf{0})$ is the predicted velocity with zero style prompt.

Table 1: Quantitative results of all methods. Time. and Mem. denote training time and GPU memory used for adaptation on a A100 GPU. StyleTalk is a one-shot method, so the Time. and Mem. are 0.

| Methods | CSIM↑ | PSNR↑ | FID↓ | AED↓ | Sync. ↑ | Time (h)↓ | Mem. (GB)↓ |
|---|---|---|---|---|---|---|---|
| RAD-NeRF (Tang et al., 2022) | 0.825 | 30.85 | 33.51 | 0.122 | 4.916 | 13.22 | **5.432** |
| GeneFace (Ye et al., 2023) | 0.819 | 30.44 | 34.16 | 0.115 | 6.480 | 16.57 | 7.981 |
| ER-NeRF (Li et al., 2023a) | 0.834 | 31.45 | 30.38 | 0.113 | 5.631 | 3.77 | 8.676 |
| StyleTalk (Ma et al., 2023) | 0.671 | 27.39 | 45.25 | 0.106 | 7.173 | / | / |
| MimicTalk (ours) | **0.837** | **31.72** | **29.94** | **0.098** | **8.072** | **0.26** | 8.239 |

Table 2: MOS score of different methods. The error bars are 95% confidence interval.

| Methods | RAD-NeRF | GeneFace | ER-NeRF | StyleTalk | **MimicTalk** |
|---|---|---|---|---|---|
| ID. Similarity | 3.96±0.29 | 3.78±0.28 | 4.06±0.24 | 3.27±0.38 | **4.15±0.20** |
| Visual Quality | 3.98±0.24 | 3.87±0.24 | 4.10±0.22 | 3.46±0.35 | **4.22±0.20** |
| Lip Sync. | 3.65±0.27 | 3.93±0.22 | 3.82±0.24 | 4.01±0.24 | **4.13±0.22** |

## 4 Experiment

### 4.1 Experimental Setup

**Implementation Details.** [2] We obtain the pre-trained person-agnostic renderer from the official implementation of Ye et al. (2024). For the SD-Hybrid adaptation, we trained the model on 1 Nvidia A100 GPU, with a batch size of 1 and total iterations of 2,000, requiring about 8 GB of GPU memory and 0.26 hours. Regarding the ICS-A2M model, we trained it on 4 Nvidia A100 GPUs, with a batch size of 20,000 mel frames per GPU. The flow-matching-based ICS-A2M model was trained for 500,000 iterations, taking 80 hours. We provide full experiment details in Appendix C.

**Data Preparation.** To evaluate the personalized renderers, we tested on ten 3-minute-long target person videos by (Tang et al., 2022) and (Ye et al., 2023). To train the ICS-A2M model, we use a large-scale lip-reading dataset, voxceleb2 (Chung et al., 2018), which consists of about 2,000 hours videos from 6,112 celebrities.

**Compared Baselines.** We compare our method with three person-dependent methods: (1) *RAD-NeRF* (Tang et al., 2022), (2) *GeneFace* (Ye et al., 2023), and (3) *ER-NeRF* (Li et al., 2023a). We also compare with a style-oriented TFG method that considers controlling the talking style, (4) StyleTalk (Ma et al., 2023). We discuss the characteristics of all test methods in Appendix A.

### 4.2 Quantitative Evaluation

We use CSIM to measure identity preservation, PSNR, FID to measure the image quality, and AED (Deng et al., 2019) and SyncNet confidence (Chung and Zisserman, 2017) to measure audio-lip synchronization. The results are shown in Table 1. We have the following observations: (1) Thanks to the powerful flow matching model and the in-context-style-mimicking ability, our method achieves the best lip accuracy (AED) and perceptual lip-sync quality (SyncNet confidence); (2) Our SD-hybrid adaptive renderer shows better visual quality than the person-specific baselines. (3) Thanks to the efficiency of the LoRA-based adaptation process, our method requires significantly less training time to adapt to a new identity within 2,000 iterations and 15 minutes (47x faster than RAD-NeRF). It also requires a low GPU memory usage (8.239 GB) for adaptation.

### 4.3 Qualitative Evaluation

#### 4.3.1 Case Study

We provide demo videos at `https://mimictalk.github.io`. We also adopted several case studies to demonstrate better performance. Specifically, (1) our *SD-Hybrid adaptation has better train-*

---

[2] We release the source code at `https://mimictalk.github.io`.

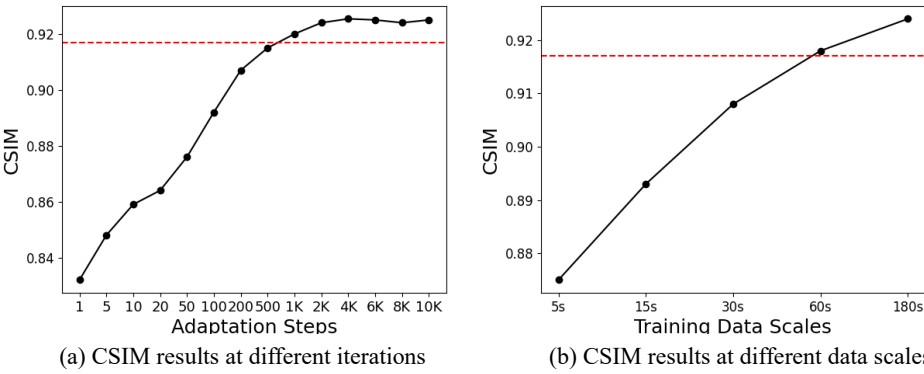

| (a) CSIM results at different iterations | (b) CSIM results at different data scales |

Figure 4: Training/data efficiency of SD-Hybrid adaptation: CSIM results at different iterations and data scales. The baseline RAD-NeRF uses 180-second-long training samples and is updated for 250,000 iterations.

Table 3: CMOS results on the style controllability and identity similarity of MimicTalk and StyleTalk. CMOS score ranges from -3 to +3. Error bars are 95% confidence intervals.

| Methods | CMOS-style-control↑ | CMOS-identitiy-similarity ↑ |
|---|---|---|
| #1. MimicTalk (ours) | $0.549 \pm 0.225$ | $1.735 \pm 0.362$ |
| #2. StyleTalk (Ma et al., 2023) | 0.000 | 0.000 |

*ing/sample efficiency than previous person-specific methods*; (2) our *ICS-A2M model predicts stylized facial motion.*

**Training / Sample Efficiency of SD-Hybrid Adaptation** To evaluate the training and sample efficiency of our SD-hybrid adaptation, we perform a case study of a talking video of President Obama provided by Tang et al. (2022). For training efficiency, as shown in Fig. 4(a), we adapt the model on a 180-second-long clip as the training data and use the lasting 10-second clip as the validation set. Our SD-hybrid adaptation enjoys a fast convergence and good performance compared to the person-specific baseline. As for the sample efficiency, we visualize the results of CSIM at different scales of training data in Fig. 4(b). It can be observed that the CSIM score improves as the amount of training data increases. Additionally, our method achieves comparable performance to the person-specific baseline, which was trained with 180 seconds of data, using only one-third of the data (60 seconds).

**Talking Style-Coherent Motion Prediction by ICS-A2M Model** Given a short reference video as the talking style prompt, our ICS-A2M model could accurately mimic its talking style (such as smiling or duck mouth). We provide a demo video at `https://mimictalk.github.io/static/videos/demo_ics_a2m.mp4` for better demonstration. We also conducted a comparative mean score opinion (CMOS) test between our method and StyleTalk (Ma et al., 2023) to qualitatively evaluate talking style accuracy. As shown in Table 3, our method achieves better talking style controllability and identity similarity. Please refer to Appendix C.3 for detailed user study settings.

### 4.3.2 User study

We conducted the Mean Opinion Score (MOS) test to evaluate the perceptual quality of generated samples, scaled from 1 to 5. Following Chen et al. (2020), the attendees are required to rate the

Table 4: Ablation studies on different settings in the SD-hybrid adptation.

| Settings | CSIM↑ | PSNR↑ | FID↓ | AED↓ | APD↓ |
|---|---|---|---|---|---|
| #1. Ours (SD-Hybrid) | **0.837** | **31.72** | **29.94** | **0.098** | **0.028** |
| #2. Ours - Tri-plane Inv. | 0.823 | 30.65 | 33.28 | 0.102 | 0.030 |
| #3. Ours - LoRAs | 0.805 | 29.95 | 36.56 | 0.108 | 0.033 |

Table 5: Ablation studies on different settings in ICS-A2M. $L2_{\text{Landmark}}$ denotes the L2 reconstruction error on the 68 3D landmarks, and $L_{\text{Sync}}$ denotes a audio-expression synchronization contrastive loss provided by (Chung and Zisserman, 2017). and (Ye et al., 2023)

| Settings | $L2_{\text{Landmark}}\downarrow$ | $L_{\text{Sync}}\downarrow$ |
|---|---|---|
| #1. Ours (ICS-A2M) | **0.026** | **0.423** |
| #2. Ours w.o. Flow Matching | 0.034 | 0.466 |
| #3. Ours w. style vec. (Huang and Belongie, 2017) | 0.031 | 0.429 |
| #4. Ours w. style enc. (Ma et al., 2023) | 0.029 | 0.428 |
| #5. Ours w.o. sync loss | 0.028 | 0.536 |

videos from three aspects: (1) *identity similarity*, (2) *visual quality*, and (3) *lip-sync*. Detailed settings are in Appendix C.3. The results are shown in Table 2. We have the following observations: (1) our method achieves the best lip-synchronization and identity similarity / visual quality compared to SOTA person-dependent methods (RAD-NeRF, GeneFace, and ER-NeRF). The MOS results demonstrate the effectiveness of the proposed MimidTalk framework.

### 4.3.3 Ablation study

**SD-Hybrid Adaptation.** We tested two settings in the SD-Hybrid adaptation: (1) Not performing tri-plane inversion to finetune a personalized tri-plane and (2) not injecting LoRAs into the model. As shown in line 2 and line 3 of Table 4, utilizing both techniques achieves the best identity similarity (CSIM), visual quality (FID), and geometry accuracy (AED and APD).

**ICS-A2M Model.** We also analyze three settings in the ICS-A2M: (1) replace the flow matching model with a deterministic transformer, as shown in line 2 of Table 5, which leads to worse motion reconstruction quality and worse sync score; (2) replace the in-context style control with a hand-crafted style vector in (Wu et al., 2021) or learn a style encoder as in StyleTalk Ma et al. (2023), as shown in line 3 and line 4, which leads to worse motion reconstruction quality, proving that the ICL talking style mimicking could prevent information loss caused by compressing the style into a global encoding; (3) remove the sync loss during training, as shown in line 5, which leads to significantly worse perceptual lip-sync performance.

## 5 Conclusion

In this paper, we propose MimicTalk, an efficient and expressive personalized talking face generation framework. We first come up with the idea of adapting a pre-trained 3D person-agnostic model to personalized datasets to inherit its generalizability and achieve fast training. The SD-Hybrid adaptation pipeline helps the generic model learn the target person's static and dynamic features, leading to better identity similarity than previous person-dependent baselines. Besides, the proposed ICS-A2M model is the first facial motion generator that enables in-context talking style control, which helps produce expressive facial motion in the generated video. Due to space limitations, we provide impact statements in Sec. E and discuss limitations and future works in Appendix D.

## 6 Acknowledgments

This work was supported in part by the National Natural Science Foundation of China under Grant No. 62222211 and National Natural Science Foundation of China under Grant No.62072397.

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

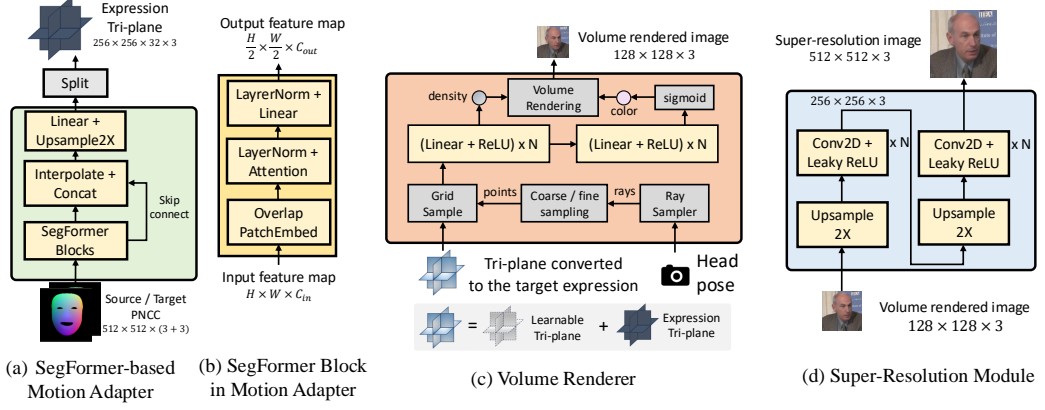

(a) SegFormer-based Motion Adapter  (b) SegFormer Block in Motion Adapter  (c) Volume Renderer  (d) Super-Resolution Module

Figure 5: The detailed network structure of the person-agnostic renderer.

# A    Comparison between Different Methods

As for the person-dependent talking face **renderer**, all previous methods like RAD-NeRF (Tang et al., 2022), GeneFace (Ye et al., 2023), and ER-NeRF (Li et al., 2023a) follow a per-identity-per-training paradigm, which means they have to learn a model from scratch for an unseen identity, which is time-consuming and the generalizability of the model is limited due to the small scale of training data. By contrast, our method is the first work to adapt a pre-trained generic one-shot 3D talking face model into the target identity. Thanks to using LoRA and the tri-plane inversion, our method could converge in 2,000 iterations (47 times faster than RAD-NeRF) and enjoy better generalizability, sample efficiency, and training efficiency due to the generic backbone.

As for the **audio-to-motion model** in the audio-driven TFG, most previous works like SadTalker (Zhang et al., 2023) and StyleTalk (Ma et al., 2023) utilize a deterministic mapping to model the audio-to-motion transform, and cannot achieve talking style control. By contrast, our method first introduces flow-matching for the audio-to-motion task and achieves in-context talking style control.

# B    Model Details

## B.1    Network Structure of the Person-Agnostic Renderer

We provide detailed network structure of the person-agnostic renderer in Figure 5.

## B.2    Details on LoRAs for SD-Hybrid Adaptation

We visualize a detailed process that plugs the LoRA into our person-agnostic renderer in Fig. 6. For each pretrained kernel or weight of shape $c_{in} \times c_{out}$, the LoRA Matrix A and LoRA Matrix B could represent a weight matrix of shape $c_{in} \times c_{out}$ and rank $r$, where $r << \min(c_{in}, c_{out})$ and we set $r = 4$ in our setting. During training, the pre-trained weights of the backbone are fixed, and only the LoRA matrices are updated.

## B.3    Details on In-Context Stylized Audio-to-Motion Model

### B.3.1    Audio-Guided Motion Infilling Task

In Fig. 7, we present a visualization of the training and inference process for the Audio-Guided Motion Infilling task. Paired audio-motion samples are required for this task, which can be easily extracted from talking face datasets and used as training data. During training, we randomly mask several segments in the motion track and encourage the model to reconstruct them based on the complete audio track and the unmasked motion context. This training approach enables the model to learn to mimic the talking style provided in the context.

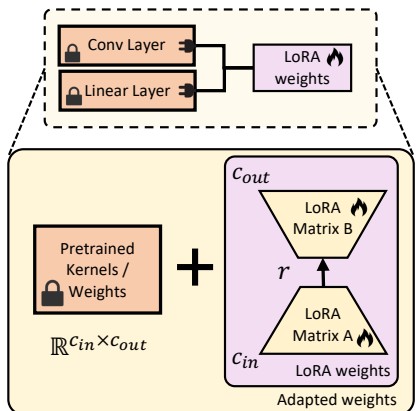

Figure 6: The process that plugs LoRAs into convolutional/linear layers of the person-agnostic renderer.

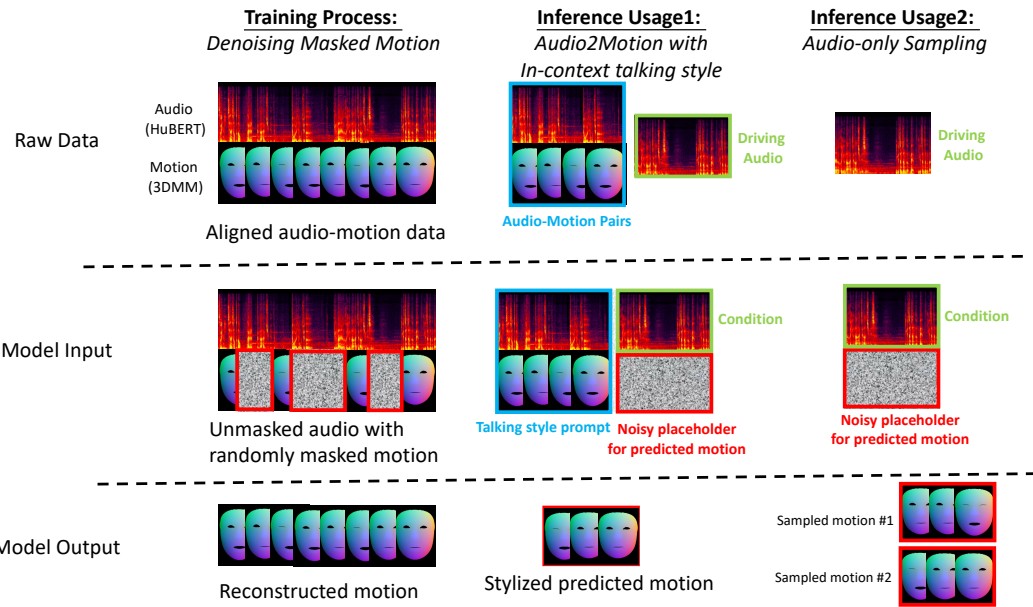

Figure 7: The training process and inference usage of the Audio-Guided Motion Infilling Task.

During inference, there are two main usage scenarios. Firstly, we can guide the model to mimic a specific talking style by providing an audio-motion pair of the target person as the talking style prompt. This setting allows for in-context talking style control over the generated facial motions. Alternatively, we also support audio-only motion sampling. Without a style prompt, the model will generate semantically correct facial motions with a randomly sampled talking style.

### B.3.2 Preliminaries on Flow Matching

In this section, we introduce preliminaries of flow matching. Conditional flow matching (CFM) is a variant of continuous normalizing flows (Chen et al., 2018), which is a class of generative models for modeling an unknown data distribution $q(x)$, where $x$ denotes the data point in the distribution. The CFM method aims to predict a time-dependent flow field that represents the velocity $u_t = \frac{dx_t}{dt}$ of the data point $x_t$ at a continuous timestep $t$, where the data point $x_0 \sim p(x)$ starts from a simple prior distribution $p(x)$ (e.g., Gaussian distribution) at the timestep $t = 0$, and is pushed towards the target distribution $q(x)$ by the velocity field as the timesteps finally approach $t = 1$. This process

can be formulated as an ordinary differential equation (ODE) as follows:

$$\frac{dx_t}{dt} = u_t \approx v_t(x_t, \theta), s.t. x_0 \sim p(x), t \in [0, 1], \tag{9}$$

where $u_t$ is the ground truth velocity for $x_t$ and is expected to be approximated by the CFM model. $v_t$ is the predicted velocity by the CFM model with the parameter $\theta$. Once the $u_t$ is obtained, we can train the model with a simple flow-matching objective:

$$\mathcal{L}_{\text{CFM}} = \mathbb{E}_t ||u_t - v_t(x_t; \theta)||_2^2. \tag{10}$$

Once the training is done, we can obtain $v_t(x_t; \theta) \approx \frac{dx_t}{dt}$, and solve the ODE in Eq. 9 to obtain the predicted sample $x_1$.

Now, we consider the specific formulation of the ground truth velocity $u_t$. We adopt the optimal transportation (OT) path proposed in (Lipman et al., 2023) to obtain the ground truth velocity. Specifically, given the target data point $x_1$ and the current data point $x_t$, the most efficient way is to go with a straight line, i.e., $u_t = (x_1 - x_t)/(1 - t)$, where $(1 - t)$ is a normalization term to make the ground truth velocity a unit vector. We choose the OT path for its simplicity and prior evidence (Lipman et al., 2023; Le et al., 2023) that it outperforms other alternatives (such as diffusion paths (Ho et al., 2020; Song et al., 2021)).

## C  Experiment Details

In the following sections, we illustrate the model configuration and training details of our MimicTalk.

### C.1  Model Configuration

We provide detailed hyper-parameter settings about the model configuration in Table 6.

Table 6: Model Configuration

|  | Hyper-parameter | Value |
|---|---|---|
| SD-Hybrid Adaptation | LoRA rank | 2 |
|  | Learnable Tri-plane shape | $256 \times 256 \times 32 \times 3$ |
| ICS-A2M Model | Transformer - Attention Hidden Size | 1024 |
|  | Transformer - Norm Type | LayerNorm |
|  | Transformer - Attention Layers | 16 |
|  | Transformer - MLP Layers per Block | 2 |
|  | Transformer - Attention Head Hidden Size | 64 |
|  | Transformer - Attention Heads | 8 |
|  | Flow Matching - Final sigma | 0. |
|  | Flow Matching - ODE method | midpoint |
|  | Flow Matching - ODE infer steps | 5 |

### C.2  Training Details

We use the pre-trained one-shot person-agnostic renderer provided in the official implementation[3] of (Ye et al., 2024).

For the SD-Hybrid adaptation, we trained the model on 1 Nvidia A100 GPU, with a batch size of 1, requiring about 8 GB of GPU memory. Surprisingly, our method achieved better results than existing person-specific baselines in just 2,000 iterations, which took about 0.26 hours and was 47 times faster than RAD-NeRF.

Regarding the ICS-A2M model, we trained it on 4 Nvidia A100 GPUs, with a batch size of 20,000 mel frames per GPU. The flow-matching-based ICS-A2M model was trained for 500,000 iterations, taking 80 hours.

---

[3]`https://github.com/yerfor/Real3DPortrait/`

## C.3 Detailed User Study Setting

As for mean score opinion (MOS) test in Table 2, we select 5 audio clips and 10 trained identities (as used in by Tang et al. (2022) and Ye et al. (2023)) to construct 50 talking portrait video samples for each method. Each video has been rated by 20 participants. We perform the identity similarity, visual quality, and lip-synchronization MOS evaluations. For MOS, each tester is asked to evaluate the subjective score of a video on a 1-5 Likert scale. For identity similarity, we tell the participants to *"only focus on the similarity between the identity in the source image and the video"*; for visual quality, we tell the participants to *"focus on the overall visual quality, including the image fidelity and smooth transition between adjacent frames"*; as for lip synchronization, we tell the participants to *"only focus on the semantic-level audio-lip synchronization, and ignores the visual quality"*.

As for comparative mean score (CMOS) test in Table 3, we first trained 10 person-specific renderers on 10 identities' ten-second-long videos. We randomly select 5 out-of-domain audio clips for driving each renderer. So there are 50 result videos for each setting. We include 20 participants in the user study. Each tester is asked to evaluate the subjective score of two paired videos on a -3 +3 Likert scale(e.g., the first video is constantly 0.0, and the second video is +3 means the tester strongly prefers the second video). To examine the aspect of (lip-sync, pose-sync, expressiveness), we tell the participants to *"only focus on the (lip-sync, pose-sync, expressiveness), and ignore the other two factors."*.

# D Limitations and Future Work

In this section, we discuss the limitations of the proposed method and how we plan to handle them in future work. Firstly, in this paper, our main focus is on the face segment. By contrast, the rigid modeling of the hair and torso segment are relatively naive and occasionally produce artifacts. We plan to adopt conditional video diffusion models like (Hu et al., 2023) to enhance the naturalness of the hair and torso segments. Secondly, we can consider more conditions, such as eyeball movement and hand gestures. Finally, the inference speed (15 FPS on 1 A100) can be improved by introducing more efficient network structures like Gaussian Splatting.

# E Broader Impacts

In this section, we discuss the ethical impacts that might be brought by the rapidly developing talking face generation technology and our measures to address these concerns.

MimicTalk facilitates efficient and expressive personalized talking face synthesis. With the development of talking face generation techniques, it is much easier to synthesize talking human portrait videos. Under appropriate usage, this technique could facilitate real-world applications like virtual idols and customer service, improving the user experience and making human life more convenient. However, the talking face generation method can be misused in deepfake-related usages, raising ethical concerns. We are highly motivated to handle these misusage problems. To this end, we plan to include several restrictions in the license of MimicTalk. Specifically,

- We will add visible watermarks to the video synthesized by MimicTalk so that the public can easily tell the fakeness of the synthesized video.

- The synthesized videos should only be used in educational or other legal usages (like online courses), and any abuse will take responsibility by tracking the method we come up with in the next point.

- We will also inject an invisible watermark into the synthesized video to store the information of the video maker so that the video maker has to account for the potential risk raised by the synthesized video.

