# OpenReview forum: "MimicTalk: Mimicking a personalized and expressive 3D talking face in minutes"
_NeurIPS.cc/2024/Conference — NeurIPS 2024 poster_

### Official Review · Reviewer_dHKC · 2024-06-20

**Soundness:** 3
**Presentation:** 3
**Contribution:** 3
**Rating:** 5
**Confidence:** 3

**Summary:**

The paper presents a novel approach namely MimicTalk to personalized talking face generation. Unlike previous methods that rely on individual neural radiance fields (NeRF) for each identity, the authors propose a more efficient and generalized framework using a person-agnostic generic model. MimicTalk introduces a static-dynamic-hybrid adaptation pipeline and an in-context stylized audio-to-motion model. The proposed method aims to achieve high-quality, expressive results while significantly reducing training time. Experimental results suggest that MimicTalk surpasses previous baselines in video quality, efficiency, and expressiveness.

**Strengths:**

1. The paper introduces a novel hybrid approach that combines static and dynamic features for personalized talking face generation. The proposed in-context stylized audio-to-motion model enhances the expressiveness of generated videos by mimicking the talking style from reference videos. The method is reasonable and interesting, and could probably bring some insights to the community.

2. The proposed method significantly reduces the training time required for generating personalized talking faces.

3. By leveraging a person-agnostic model, the paper demonstrates improved robustness and efficiency in handling various identities and out-of-domain conditions.

4. The authors promised to release the code.

**Weaknesses:**

1. The paper claims superior performance over existing methods, but the experimental results presented are not sufficiently comprehensive or detailed to fully substantiate these claims. Although some video results are provided in the URL, the results are not convincing enough. In particular, I cannot see the advantages of the proposed method in the style-control results. There are also not enough examples for comparison to state-of-the-art approaches.

2. The experiments are limited in scope, primarily focusing on a narrow set of scenarios and datasets. The evaluation lacks a diverse range of conditions such as varying lighting, occlusions, and different levels of head movements, which are crucial for real-world applicability. More comprehensive testing across varied and challenging scenarios would strengthen the validity of the claims.

**Questions:**

See weaknesses section.

**Limitations:**

The paper addressed the limitations in the appendices.

---

> ### Author Rebuttal · Authors · 2024-08-06
>
> We are grateful for your positive review and valuable comments, and we hope our response fully resolves your concerns.
>
> > Q1: The experimental results presented are not sufficiently comprehensive or detailed to fully substantiate these claims. Although some video results are provided in the URL, the results are not convincing enough. In particular, I cannot see the advantages of the proposed method in the style-control results.
> - A1: Thanks for your helpful feedback. We have enriched the results of talking style control. Specifically, 1) we tried different classifier-free guidance (CFG) scales in the sampling process of the ICS-A2M model to further improve the talking style control. We provide a new demo video (please kindly refer to **Rebuttal Demo 4** on the demo page) with 6 additional style control examples to help the reader make a qualitative comparison. 2) We also perform user studies to evaluate the style control ability of our method and the baseline StyleTalk. Please refer to **Table 2 in the attached one-page PDF** for details. Both qualitative/quantitative evaluations show that our method performs better in talking style control.
>
> > Q2: There are also not enough examples for comparison to state-of-the-art approaches.
> - A2: We acknowledge that making comparisons on more identities with diverse languages and appearances could better prove the generalizability of our method. To this end, we provide an additional demo video (please kindly refer to **Rebuttal Demo 1** on the demo page) with 9 additional identities and the results prove that our method has better data efficiency and lip-sync quality than the most competitive baseline ER-NeRF (ICCV 2023).
>
> > Q3: The experiments are limited in scope. The evaluation lacks a diverse range of conditions such as different levels of head movements, which are crucial for real-world applicability.
> - A3: Thanks for your insightful comment! We admit that the limited evaluation scope is a common problem in the field of NeRF-based talking face generation. And we have performed an additional experiment that drives our method and the baseline with out-of-domain (OOD) head movements.  We provide an additional demo video (please kindly refer to **Rebuttal Demo 3** on the demo page), and the results show that our method could well handle the OOD head movement while the baseline cannot. We suspect the prior knowledge in our generic backbone is the reason for our method's generalizability to OOD poses. We will add this experiment to the revised manuscript. Besides, as for other critical situations, as can be seen **Rebuttal Demo 1** on our demo page, our method is more robust than the baseline for difficult training videos (such as identities with long hair or sided faces). We also plan to benchmark our method on more critical datasets such as varying lighting and occlusions in the future.
>
> # Summary
> Following your comments, we have performed several additional experiments and found the experiment results have been more comprehensive and convincing. Again, we thank the reviewer for the insightful review and positive recommendation for our paper.

---

> ### Author Response · Authors · 2024-08-13
> **Hoping that our response could address your concern**
>
> Dear Reviewer dHKC,
>
> Thank you again for your time and effort in reviewing our work! We would appreciate it if you can let us know if our response has addressed your concern. As the end of the rebuttal phase is approaching, we look forward to hearing from you and remain at your disposal for any further clarification that you might require.

---

> ### Author Response · Authors · 2024-08-14
> **Dear Reviewer**
>
> Dear Reviewer dHKC.
>
> As the discussion period is closing in several hours, we would like to know if there are any additional questions. We are glad to answer them.
>
> Again, we sincerely appreciate your insightful review and positive recommendation for our paper.

---

### Official Review · Reviewer_TKxi · 2024-07-01

**Soundness:** 3
**Presentation:** 3
**Contribution:** 3
**Rating:** 5
**Confidence:** 5

**Summary:**

The paper presents MimicTalk, an approach to improve the efficiency and robustness of personalized talking face generation. Instead of using separate NeRFs for each identity, MimicTalk adapts a person-agnostic NeRF-based model for specific individuals. They also propose an in-context stylized audio-to-motion (ICS-A2M) model to create facial movement that imitates the target person's speaking style. The adaptation process converges quickly for unseen identities. Experiments show that MimicTalk outperforms previous baselines in terms of video quality, efficiency, and expressiveness.

**Strengths:**

1.  This paper can achieve fast adaption from generic to person-specific model, and the overall reconstruction quality seems good compared to other methods.
2. Several proposed improvement also has been proved as effective.

**Weaknesses:**

1.  The head pose of the talking face video is not generated, for a talking face video, the sync of the head pose and audio is very important.
2. The facial motion is represented by PNCC, which limits the expressiveness of the generated results, e.g. detailed emotions, eyeball movements.

**Questions:**

1.  The $L_{sync}$ provided in **GeneFace** is used to evaluate the synchronization between audio and 68 sparse landmarks. In this paper, the authors call it audio-expression synchronization loss, which is not adequate. Sparse landmarks cannot reflect the richness and expressiveness of facial motions. It seems the judgement of the motion expressiveness (FID/user study) is missing in this paper.
2.  The results of talking style control are not sufficient. Only two examples are given. More qualitative/quantitative evaluations of the style controllability are needed.
3. Since this is about 3D talking face generation. It would be better to see some novel-view results.

**Limitations:**

The authors have discussed the limitations and broader impacts in their appendix. The authors should also discuss the limitations as I list in the **Weaknesses** section.

---

> ### Author Rebuttal · Authors · 2024-08-06
>
> We thank the reviewer for the constructive feedback and the positive remarks on our Soundness and Contribution. We acknowledge that your concerns are mainly about the experiment of this paper, and hope our response resolves your concerns fully.
>
> > Q1: The head pose of the talking face video is not generated.
> - A1: Thanks for pointing this out! In the original paper, we didn't consider predicting the head pose since we needed to calculate error-based metrics such as average pose distance (APD). We acknowledge that the sync of the head pose and audio is important for talking head generation. Therefore, we additionally train an **audio-to-pose model**, which follows the main structure of the ICS-A2M model proposed in the original paper. Please kindly refer to the **Rebuttal Demo Video 2** on our demo page, whose URL is given in the original paper (due to the rebuttal policy). The Rebuttal Demo 2 shows that our model can produce novel head poses that are coherent with the input audio. We will add the audio-to-pose model to the revised manuscript. Thanks for your helpful comment!
>
> > Q2: The motion representation PNCC may limit the expressiveness of the generated results, such as detailed emotions or eyeball movements.
> - A2: We acknowledge that the PNCC cannot represent subtle emotion and eyeball movement. This is a common problem for all talking face methods that use explicit motion representations (e.g. 3DMM exp code, landmark, PNCC). In the future, we will explore implicit representation such as audio-to-latent to improve the expressiveness of the model. We will add the above discussion to the limitation section in the revised version. Thanks for your insightful suggestion!
>
> > Q3: Sparse landmark-based $L_{sync}$ metric cannot reflect the richness and expressiveness of facial motions. It seems the judgment of motion expressiveness (FID/user study) is missing in this paper.
> - A3: We acknowledge that sparse landmarks (used in GeneFace) or the 3DMM expression code (used in this paper) can cause information loss in measuring the expressiveness of facial motion, so the audio-expression sync loss $L_{sync}$ is not a good choice for evaluation metrics in Table 5 of the original paper. We found it non-trivial to design an objective metric for audio-lip expressiveness. Following your suggestion, we turned to user study. Specifically, we adopt the Comparative Mean Opinion Score (CMOS) as the metric of lip-sync (CMOS-sync) and expressiveness (CMOS-expressive). Please kindly refer to **Table 1 in the attached one-page PDF** for details. We can see that our default setting performs best and the other 4 settings or the baseline ER-NeRF lead to lower CMOS scores.
>
> > Q4:  The results of talking style control are not sufficient. More qualitative/quantitative evaluations of the style controllability are needed.
> - A4: Thanks for your helpful feedback. We have made amendments to the results of talking style control. Specifically, 1) we provide an additional demo video (please kindly refer to **Rebuttal Demo 4** on the demo page) with more style control examples to help the reader make a qualitative comparison. 2) We additionally perform user studies (the Comparative Mean Opinion Score test) to evaluate the style control ability of our method and the baseline StyleTalk. Please refer to **Table 2 in the attached one-page PDF** for details. Both qualitative/quantitative evaluations show that our method performs better in talking style control.
>
> > Q5:  It would be better to see some novel-view results.
> - A5: Thanks for your suggestion! We provide an additional demo video (please kindly refer to **Rebuttal Demo 3** on the demo page), in which we drive our method and the baseline with several out-of-domain poses. We found that our method can well handle the OOD poses and generate high-quality results while the baseline cannot. Possibly, the prior knowledge in our generic backbone is the reason for the generalizability of OOD poses.
>
> # Summary
> In summary, following the given comments, we have performed in-depth analysis and several experiments, which we find has enhanced the soundness of the paper and proved the performance improvement of the proposed method. Again, we would like to appreciate the reviewer’s valuable review. We sincerely hope the reviewer will reconsider their rating in light of the rebuttal.

---

> > ### Comment · Reviewer_TKxi · 2024-08-09
> > **Missing details for rebuttal response**
> >
> > Thank you for your response. However, I still have some questions and concerns as follows:
> >
> > 1. Why do you model audio-to-pose and audio-to-face motion separately? Since head pose should align with face motion, it would be better to model them together. I understand the rebuttal time is limited, but if you want to claim to handle head pose generation, you should include some evaluations in your revised version.
> >
> > 2. Could you provide more details about the user study? e.g., how many individuals participated and how many cases were evaluated?
> >
> > 3. I checked **Rebuttal Demo 4** for more style prompts. The results aren't good. A larger CFG sometimes enhances the style's intensity, but it also adds more artifacts. These results don't seem to support the claim of style mimicking.

---

> ### Author Response · Authors · 2024-08-10
> **Author Response to Reviewer TKxi (Part 1/2)**
>
> Dear Reviewer yTCz,
>
> Thanks for your fast response! Sorry for the late reply since we were doing the relevant additional experiments. We hope our response fully resolves your remaining concerns.
>
> > Q1:  It would be better to model the audio-to-motion and audio-to-pose together. Please include some evaluations of the audio-to-pose in your revised version.
> - Thanks for your feedback. In the initial rebuttal, we train a new audio-to-pose model from scratch due to its simplicity. Following your suggestion, we tried to **jointly model the audio-to-pose and audio-to-face motion together** . Specifically, we load the pre-trained audio-to-motion model and change the output dimension of its final layer: from the 64 dimensions (3DMM expression) to 70 (64+6) dimensions, in which the  6 additional dimensions denote the predicted head pose, including Euler angle (3 dim) and head translation (3 dim). We finetuned the audio-to-motion/pose model for 50,000 steps until reaching its convergence.
> - We then perform a **quantitative evaluation to evaluate audio-lip/pose synchronization and expressiveness** in different settings of the audio-to-motion/pose model. Specifically, we finetune 10 person-specific renderers on 10 ten-second-long training videos. Then we use the different settings of the generic audio-to-motion/pose model to predict the face motion or head pose. To be coherent with real-world applications, we randomly select 5 out-of-domain audio clips for driving each renderer. So there are 50 result videos for each setting. Since there is no ground truth for the result videos, we turn to user study for evaluation. We included 20 attendees and asked them to evaluate the comparative mean opinion score between each setting with the results of our original version. The attendees are required to rate in terms of audio-lip-sync, audio-pose-sync, and overall expressiveness. Please refer to **Q2 & A2 for more details of the user study**. The results are shown as follows. The error bars are 95\% confidence interval.
> | settings      | CMOS audio-lip-sync | CMOS audio-pose-sync     | CMOS-expressive  |
> | :---        |    :----:   |             :----:   |      :----: |
> | Audio-to-Motion + poses extracted from other videos （original version）      | 0.000       | 0.000   | 0.000   |
> | Audio-to-Motion + Audio-to-Pose, separately (initial rebuttal version)   | $0.215 \pm 0.293$        | $1.341 \pm 0.284$      |  $0.647 \pm 0.251$   |
> | Audio-to-Motion/Pose, jointly (current version)  | $0.468 \pm 0.235$        | $1.653 \pm 0.316$      |  $0.952 \pm 0.301$   |
>
> - In line 1 and line 2, we can see that predicting the pose from audio greatly improves the audio-pose-sync performance, and in line 3 we can see that jointly predicting the face motion and head pose could further improve the performance from audio-lip-sync, audio-pose-sync, and expressiveness. We suppose that it is because the head pose strictly aligns with face motion, and there is a high correlation between them (facial motion can be a strong hint for predicting the head pose and vice versa). Therefore, joint modeling of motion and pose could improve the model's ability to better predict them. We found the additional modeling of head pose greatly improves the naturalness of our method, and will add the above discussion in the revised manuscript. Thanks for your insightful suggestion!
>
> > Q2: Could you provide more details about the user study? e.g., how many individuals participated and how many cases were evaluated?
>
> A2: In the additional user study during the rebuttal, we follow the main setting as we did in the paper. Specifically, we first trained 10 person-specific renderers on 10 identities' ten-second-long videos. We randomly select 5 out-of-domain audio clips for driving each renderer. So there are **50 result videos for each setting**. We include **20 participants in the user study**. The biggest difference is that we test the **Comparative Mean Opinion Score (CMOS)** instead of the Mean Opinion Score (MOS) in the original paper to better evaluate the comparative performance between different settings. For CMOS, each tester is asked to evaluate the subjective score of two paired videos on a -3~+3 Likert scale(e.g., the first video is constantly 0.0, and the second video is +3 means the tester strongly prefers the second video). To examine the aspect of (lip-sync, pose-sync, expressiveness), we tell the participants to *"only focus on the (lip-sync, pose-sync, expressiveness), and ignore the other two factors."*
>
> Due to space limitations, please refer to the next comment for the **reply to Q3**.

---

> ### Author Response · Authors · 2024-08-10
> **Author Response to Reviewer TKxi (Part 2/2)**
>
> > Q3: I checked Rebuttal Demo 4 for more style prompts. The results aren't good. A larger CFG sometimes enhances the style's intensity, but it also adds more artifacts. These results don't seem to support the claim of style mimicking.
> - We have provided an additional **Post-Rebuttal Demo 1**. Please kindly refer to our demo page for a quick view.
> - Thanks for your feedback. Firstly we want to **clarify the claim of style mimicking**. We acknowledge that the claim of style mimicking in the initial manuscript is somewhat misleading. Note that the task of this paper is personalized talking face generation (TFG), in which we have a short training video clip of the target identity, and we aim to train a TFG system that mimics not only its visual attributes but also its talking style. Therefore, in personalized TFG, we only need to run our model with the in-domain style prompt of the target identity (**in-domain style prompt** denotes using the training video clip as the style prompt). Actually, in Lines 74-78 of the original manuscript, we claim that the initial intention of proposing the ICS-A2M model is to let the model in-context (i.e., without the need for finetune) mimic the target identity's talking style. By contrast, as for supporting out-of-domain style prompts (**OOD style prompt** denotes cross-identity videos that have critical expressions like duck face that are unseen in the training video of the target identity), we regard it as an interesting feature of our method, rather than the key problem to be solved in this paper. As shown in **Post-Rebuttal Demo 1**, using the target identity's in-domain style prompt, our method could produce expressive results with good identity similarity and style similarity, and the performance is robust to a large CFG scale (e.g, cfg_scale=4). This result demonstrates that our method achieves the goal of personalized TFG (i.e., mimicking the target identity's visual attributes and talking style). We feel sorry for causing the misunderstanding, and **we will revise the claim of style mimicking** in the original manuscript by adopting the above discussion into the Introduction Part. We will emphasize that the major goal of the proposed ICS-A2M model is to better mimic the talking style of the target identity to achieve personalized TFG, and **we will introduce OOD talking style mimicking as an interesting feature** of our model in the Experiment Part.
> - Then we want to discuss **why the OOD style mimicking is not very robust**. In the Rebuttal Demo 4, we used various OOD style prompts to drive the target identities (e.g., Obama never played a duck face in the training video clip, but we set Trump's duck face video clip as the style prompt of the Obama renderer). Considering the scarce of motion-image pair in the ten-second-long training video, it is hard to render OOD extreme expressions without artifacts (e.g., for a model trained on a video that the speaker never says "oh", it is difficult to synthesize the speaker saying "oh" with good image quality). To analyze the reason for the artifact, we visualize the facial motion generated by our ICS-A2M model at various CFG scales and notice that the **ICS-A2M model could faithfully produce lip-sync facial motion of OOD talking styles at all tested CFG scales**. Now that the audio-to-motion stage works well, we suspect that the reason behind the visual artifact is that, when increasing the CFG scale, the generated facial motion is getting more similar to the OOD style prompt. However, the facial motion pattern in the OOD style prompt is quite different from the training motion condition of our person-specific renderer, hence resulting in visual artifacts.
> - To **improve the robustness of OOD style mimicking**, we found two possible solutions: **1)** the first is to **tune the CFG scale** as a hyper-parameter. For instance, in Rebuttal Demo 4, while CFG=4 occasionally leads to artifacts, CFG=2 is a stable choice that balances the style similarity and visual quality. **2)** the second direction is to improve the generalizability of the motion-conditioned renderer. We could adopt **data augmentation** to synthesize images of diverse facial motions through recent advanced face editing methods, which can improve OOD performance, and hence could facilitate more stable OOD style mimicking. Due to time limitations, we leave this as future work.
>
> Finally, we'd like to thank you for your precious time and valuable comments, which have improved the soundness and clarity of this manuscript. We will be very happy to clarify any remaining points (if any).

---

> ### Comment · Reviewer_TKxi · 2024-08-12
> **Final Rating**
>
> After reading your response, most of my questions have been addressed. However, I believe that compared to the submitted version, many changes are needed. For example, the user studies, the experiments on pose generation, and the claims regarding the contribution of style mimicking should be revised. I have mixed feelings about this. I improve the rating, but I am more inclined to suggest resubmitting to another conference to refine these issues. The final decision can be left to the area chair.

---

> > ### Author Response · Authors · 2024-08-12
> > **Thanks for your acknowledgment of our rebuttal**
> >
> > Dear Reviewer TKxi,
> >
> > Thanks for your engaged discussion and acknowledgment of our rebuttal. We'd like to especially thank you for the three suggestions (details of user study, head pose prediction experiments, and clarifying the contribution of style mimicking). We have integrated the latest discussion and additional experiments into the revised manuscript, which we found has improved the soundness and completeness of the paper.
> >
> > Again, thank you for your expertise and positive feedback!

---

### Official Review · Reviewer_iWSN · 2024-07-06

**Soundness:** 4
**Presentation:** 3
**Contribution:** 3
**Rating:** 7
**Confidence:** 4

**Summary:**

This MimicTalk work aims to bridge the gap between the person-agnostic one-shot TFG setting and the person-dependent small-scale TFG setting. A carefully designed static-dynamic-hybrid adaptation pipeline is proposed to achieve expressive, generalized, and efficient personalized TFG. Furthermore, an in-context stylized audio-to-motion model is introduced to mimic the talking style from the reference video. Experiments demonstrate the superiority of MimicTalk compared to previous methods.

**Strengths:**

The introduction part presents a good observation of identity-dependent and identity-agnostic methods. To overcome their limitations, a static-dynamic-hybrid adaptation pipeline is introduced to first build an initial 3D face mode from a pretrained one and then finetune it with small-scale person-specific video data. To make the finetune process more efficient and stable, the LoRA technique is adopted. The design motivation is clear and intuitive.

To achieve audio-to-motion generation, an In-Context Stylized Audio-to-Motion module is introduced built upon flow matching, and trained via infilling mask task.

The illustrated video results and quantitative results show the effectiveness and efficiency of the introduced pipeline.

The paper is well-organized and easy to follow. The illustration and implementation details are well demonstrated.

**Weaknesses:**

There are several losses used in the pipeline. How to choose these loss weights and how these loss weights affect the performance could be further discussed.

**Questions:**

The loss weight choice could be discussed.

**Limitations:**

The limitations have been discussed.

---

> ### Author Rebuttal · Authors · 2024-08-06
>
> We are grateful for your positive review and valuable comments, and we hope our response fully resolves your concerns.
>
> > Q1: There are several losses used in the pipeline. How to choose these loss weights and how these loss weights affect the performance could be further discussed.
>
> - A1: Thanks for your advice. Since the training objective of the renderer and the ICS-A2M consists of multiple terms, we acknowledge that there should be more discussion on choosing the appropriate loss weights to achieve a good overall quality. Specifically,
> - 1\) as shown in Equation (4) of the original paper, the loss of the renderer is:
>
> $$\mathcal{L}_\text{renderer} = \mathcal{L}_1 + \lambda _{LPIPS}  \cdot \mathcal{L} _\text{LPIPS}  + \lambda _{ID} \cdot \mathcal{L} _\text{ID} .$$
> - Where $\mathcal{L}_ 1$ is the L1 loss between the predicted frame and the GT frame and is the main objective. However, using L1 loss alone is known to cause over-smoothing in the generated sample, hence we additionally adopt LPIPS loss to improve the high-frequency texture details in the generated result. However, LPIPS loss is insensitive to tiny spatial offsets, hence we take LPIPS as an auxiliary loss to improve image quality and set a relatively small loss weight $\lambda _\text{LPIPS}=0.2$. Identity loss $\mathcal{L} _{ID}$ is similar to LPIPS loss, except that the loss is empowered by VGGFace, which focuses more on face similarity. We find that a large identity loss causes training instability, hence setting a small loss weight $\lambda _\text{ID}=0.1$.
>
> - 2\) As for the ICS-A2M model, as shown in Equation (6) of the original paper, the loss is:
>
>  $$\mathcal{L} _\text{ICS-A2M} = \mathcal{L} _\text{CFM} + \lambda _\text{sync}\cdot\mathcal{L} _\text{sync},$$
>
> - where L_CFM is the conditional flow matching (CFM) objective and L_sync is a discriminative loss that measures the synchronization between the input audio and the predicted expression. We found sync loss is necessary to improve lip-sync quality as otherwise the generated motion tends to be over-smoothed. However, a large weight of sync loss (e.g.) makes the generated sample over-flicking, and we found $\lambda_\text{sync}=0.05$ is a good choice to balance the temporal stability and lip-sync accuracy of the generated sample.
> - We plan to add the above discussion in Appendix B. of the revised manuscript, which we found could help the reader to understand the design of the proposed multi-objective training loss.
>
> # Summary
> Following your comments, we did an in-depth analysis of the design of the proposed method, which we found has improved the clarity of the paper. Again, we thank the reviewer for the insightful review and "Accept" recommendation for our paper.

---

> ### Comment · Reviewer_iWSN · 2024-08-10
> **Thanks for the detailed rebuttal**
>
> I have read other reviews and author response. My concers have been addressed. The additional experiments on OOD poses and the limitation discussion of PNCC (raised by Reviewer TKxi) are interesting points to explore. I suggest including these discussions and results in the final version.

---

> > ### Author Response · Authors · 2024-08-10
> > **Author Response to Reviewer iWSN**
> >
> > Dear Reviewer iWSN,
> >
> > Thanks for your acknowledgment of our rebuttal. We will add the discussion on the design of the multi-objective loss functions of our model in the Method Section. We will also add the additional experiments on OOD poses and the limitation discussion of PNCC in the revised manuscript.
> >
> > Again, we'd like to thank you for your precious time, expertise, and deep understanding of our work.

---

### Official Review · Reviewer_fWzj · 2024-07-14

**Soundness:** 2
**Presentation:** 2
**Contribution:** 2
**Rating:** 5
**Confidence:** 3

**Summary:**

This paper targets to tackle efficient 3D realistic talking face customization. Rather than learning an individual neural radiance field for each identity, this work exploits a person-agnostic model to improve the efficiency and robustness of personalized talking face generation. A static-dynamic-hybrid adaptation pipeline is proposed to adapt a person agnostic base model to a specific identity. To achieve style control, an in-context audio-to-motion model is devised to mimic the talking style of reference video.

**Strengths:**

1. Overall, I like the idea of first learning a generic base talking face model, on top of which the model is adapted to a personalized identity.

2. The design of adapting the neural volume through LoRA with PNCC input is intuitive and interesting.

3. To generate personalized facial motion, an in-context style mimicking audio-driven module is proposed to inject the dynamic motion style.

**Weaknesses:**

1. According to the attached supplementary video, the improvement seems very marginal or even hard to notice. Also, only three personal identities are showcased. It will be more persuasive if more identities are demonstrated. I am concerned whether the PNCC representation offers sufficient information or not. At the same time, the simultaneous optimization of a learnable reconstructed face and LoRA looks weird, making it confusing about which component is really functioning.

2. Authors claim this approach is a fast adaptation method that significantly surpasses the previous person-dependent approaches. In the comparison approaches, all the 3 Nerf based methods do not aim to improve the efficiency. Is there any related work that also handles the efficiency of generation or this is the first work. If not, authors should also include relevant approaches in this paper. Specifically, there might be other works targeted to efficiently fitting to a specific identity rather than adaptation. Will it be better to also take them into consideration instead of making a strong claim of 47 times speed faster.

3. The information in the workflow of Fig.2 is minimal. The representation of the NerF field uses a cube, which might have a clearer description. Meanwhile, each component of the devised network is drawn as a block diagram, which might require clearer description on the specific architecture details (If not here, the appendix). Similarly, for the inference pipeline, authors spent a lot of effort to illustrate the flow-matching training paradigm, which makes this figure a little bit messy.

**Questions:**

Please refer the weakness section.

**Limitations:**

Authors adequately addressed the limitations.

---

> ### Author Rebuttal · Authors · 2024-08-06
>
> # Author Response to Reviewer fWzj (Part 1/2)
>
> We thank the reviewer for the constructive feedback and the positive remarks on our proposed "generic-model-to-adaptation" framework. We acknowledge that your concerns are mainly about qualitative results and some technical designs, and hope our response resolves your concerns fully.
>
> > Q1: The improvement seems very marginal or even hard to notice. Only three personal identities are showcased. It will be more persuasive if more identities are demonstrated.
> - A1: Thanks for your helpful feedback. To better compare the performance, we additionally tested 9 additional identities (from an AAAI 2024 Kaggle competition of talking face generation) with diverse languages and appearances and critical conditions (such as large head pose or long hair). Please kindly refer to **Rebuttal Demo 1** on our demo page (please refer to the paper for the URL) and the results prove that our method has better data efficiency and lip-sync quality than the most competitive baseline ER-NeRF (ICCV 2023). Besides, in the **Rebuttal Demo 3** on our demo page, we show that our method could well handle the OOD pose while the baseline cannot. We suspect the prior knowledge in our generic backbone is the reason for our method's generalizability to OOD poses.
>
> > Q2: I am concerned whether the PNCC representation offers sufficient information or not.
> - A2: PNCC is a rasterized image obtained by projecting the 3DMM mesh of the target identity. It possesses fine-grained geometry information of the human head of the target expression. So we suspect the PNCC's amount of information is equal to (or slightly less than) the 3DMM mesh. There are several previous works that also adopt PNCC and achieve subtle expression control. We also provide an additional demo video (**Rebuttal Demo 4 on our demo page**), in which we show that using PNCC representation we can control various talking styles.
> - However, we acknowledge that the PNCC (or other explicit motion representations like 3DMM expression code or facial landmark) makes it hard to represent subtle movements like eyeball movement, which is a common problem for all talking face methods that use explicit motion representations. We plan to explore implicit representation such as audio-to-latent to improve the expressiveness of the model. We will add the above discussion to the limitation and future work section in the revised version. Thanks for your insightful suggestion!
>
> > Q3:  The simultaneous optimization of a learnable reconstructed face and LoRA looks weird, making it confusing about which component is really functioning.
> - A3: Thanks for your feedback. As shown in **line 2 and line 3 of Table 3 in the original paper**, only optimizing either the learnable reconstructed face or LoRA leads to a significant performance drop. Actually, the simultaneous optimization of these two components is the reason for the name of our adaptation pipeline "static-dynamic-hybrid adaptation" in Section 3.3. Specifically, learning the reconstructed face (i.e., the triplane representation) is to optimize the target identity's static attributes (such as geometry shape or appearance), and optimizing the LoRA in the model backbone is to learn the personalized dynamic attributes (such as how the facial muscle moves when doing a specific expression). **Empirically**, we find only optimizing the learnable reconstructed face could preserve the static details like teeth and hair of the target identity, but the talking avatar lacks personalized dynamic attributes of the target person (e.g., Theresa May has deep wrinkles when smiling). On the other hand, only optimizing the LoRA could produce videos with personalized facial subtle motions, but the static details are missing (e.g., high-frequency in the hair or teeth region). Therefore, we find a joint optimization of the learnable reconstructed face and LoRA could simultaneously learn the static and dynamic attributes of the target identity.
> - We found our original discussion on the design of SD-hybrid adaptation is insufficient and may cause confusion. We will add the discussion above in Section 3.3 of the revised version. Thanks for your helpful comment!
>
> > Q4: Is there any related work that also handles the efficiency of generation or this is the first work. If not, the authors should also include relevant approaches in this paper. Specifically, there might be other works targeted at efficiently fitting a specific identity rather than adaptation. Will it be better to also take them into consideration instead of making a strong claim of 47 times speed faster?
> - A4: Thanks for pointing this out! We notice that there has been training acceleration from the earliest NeRF-based AD-NeRF (ICCV 2021, taking 40 hours for training an identity) to RAD-NeRF (arxiv 2022, taking 10 hours) to the recent ER-NERF (ICCV 2023, taking 4 hours). However, this improvement of efficient fitting is caused by the improvement of model structure (from the AD-NeRF's MLP-based vanilla NeRF to the RAD-NeRF's grid-based instant-NGP, to the ER-NeRF's attention-enhanced instant-NGP). To our knowledge, we are the first work that improves training efficiency by proposing a new training paradigm (fast adapting from a pre-trained person-agnostic model), which is orthogonal to previous works that improve the network structure. By exploiting the rich prior knowledge from the generic model backbone, our method achieves not only faster convergence but also better video quality and generalizability. We hope the proposed "generic-model-to-adaptation" training paradigm could pave the way for a new generation of NeRF-based works. We will add the discussion above in the related works section of the revised version, which we find could better highlight the novelty of this paper.

---

> ### Author Response · Authors · 2024-08-06
> **Rebuttal by Authors**
>
> # Author Response to Reviewer fWzj (Part 2/2)
>
> > Q5: The representation of the NeRF field needs a clearer description. Meanwhile, each component of the devised network is drawn as a block diagram, which might require a clearer description of the specific architectural details.
> - A5: Thanks for your suggestion. We use tri-plane representation to represent the 3D NeRF field. We have replaced the cube with a tri-plane image and added a label description to improve the clarity of Figure 2 in the original paper. Please refer to **Figure 1 in the attached one-page PDF** for details. As for the specific architectural details of each block diagram, we provide an additional figure that plots the architectural details of each component in our generic model. Please refer to **Figure 2 in the attached one-page PDF** for details. Actually, in the original paper, we omitted the structure details because we want to imply that our SD-Hybrid method can be applied to an arbitrary one-shot person-agnostic model, not only the model we used in the paper. However, we acknowledge that it is necessary to provide network details to make it easier for readers to understand the role of each component in the model. We will add Figure 2 in the attached one-page PDF in Appendix B.1 of the revised manuscript.
>
> > Q6: For the inference pipeline (Fig 3 in the original paper), the authors spent a lot of effort to illustrate the flow-matching training paradigm, which makes this figure a little bit messy.
> - Thanks for pointing it out. In Figure 3 of the original paper, we introduced the model input and the sample processes of the ICS-A2M model, respectively. The intention is to highlight two novelty points of our ICS-A2M model: The first is the masked input that concatenates the style prompt with the audio condition, which is the key contribution that enables in-context talking style control; The other is the flow-matching sample process that predicts velocity. Note that we are the first to use flow-matching for modeling audio-to-motion mapping. However, we acknowledge that this diagram is a bit messy. We provide a revised version of this figure to improve clarity. Please refer to **Figure 3 in the attached one-page PDF**.
>
> # Summary
> In summary, following the given comments, we have performed several experiments and revised the manuscript from several aspects, which we find has enhanced the soundness and clarity of the paper. Again, we would like to appreciate the reviewer’s valuable review. We sincerely hope the reviewer will reconsider their rating in light of the rebuttal.

---

> ### Author Response · Authors · 2024-08-13
> **Hoping that our response could address your concern**
>
> Dear Reviewer fWzj,
>
> Thank you again for your time and effort in reviewing our work! We would appreciate it if you can let us know if our response has addressed your concern. As the end of the rebuttal phase is approaching, we look forward to hearing from you and remain at your disposal for any further clarification that you might require.

---

> ### Author Response · Authors · 2024-08-14
> **Dear Reviewer**
>
> Dear Reviewer fWzj.
>
> As the discussion period is closing in several hours, we would like to know your feedback on our rebuttal and if there are any additional questions. We are glad to answer them.

---

### Author Rebuttal · Authors · 2024-08-06

# General Response
We would like to thank the reviewers for their constructive reviews! Following the comments and suggestions of reviews, we have performed additional experiments and revised the manuscript. We have also uploaded **4 new demo videos on the demo page** (Please kindly refer to our original paper for the URL due to the rebuttal policy). Here we summarize the revision as follows:
- As suggested by Reviewer fWzj and dHKC, we provide a demo video (**Rebuttal Demo 1** on the demo page) that compares our method with the baseline of 9 additional identities, and the results prove that our method has better data efficiency and lip-sync quality.
- To predict head pose from audio, as suggested by Reviewer TKxi, we additionally train an audio-to-pose model, which follows the main structure of the ICS-A2M model proposed in the original paper. As shown in **Rebuttal Demo 2** on the demo page, our model can produce novel head poses that are coherent with the input audio.
- As suggested by Reviewer TKxi and dHKC, we provide a demo video (**Rebuttal Demo 3** on the demo page) that compares our method with the baseline when driven by various OOD head poses, and the results show that our method could well handle the OOD pose while the baseline cannot.
- As suggested by Reviewer dHKC and fWzj, we provide a demo video (**Rebuttal Demo 4** on the demo page) that tests the talking style mimicking ability of our method. By tuning the classifier-free guidance (CFG) scale during the sampling process of the ICS-A2M model, we further improve the style similarity for our flow-matching-based model. The results show that our method could handle various style references well (6 prompts in the video).
- As suggested by Reviewer iWSN and fWzj, we have enriched technical content such as how to make tradeoffs among multiple losses, discussion on previous works that also explore training-efficient NeRF-based talking face generation, and improved the clarity of Figure 2/3 in the original paper. Please refer to the **attached one-page PDF** for details.

Thanks again for the reviewers' great efforts and valuable comments, which have improved the soundness of the manuscript. We have carefully addressed the main concerns and provided detailed responses to each reviewer. We hope you will find the responses satisfactory. We would be grateful if we could hear your feedback regarding our answers to the reviews.

---

### Author Response · Authors · 2024-08-12
**Dear AC and Reviewers,**

Thanks again for your great efforts and valuable comments.

We have carefully addressed the main concerns and provided detailed responses to each reviewer. We hope you might find the responses satisfactory. As the end of the rebuttal phase is approaching, we would be grateful if we could hear your feedback regarding our answers to the reviews. We will be very happy to clarify any remaining points (if any).

---

### Decision · Program_Chairs · 2024-09-25

**Decision:**

Accept (poster)

**Comment:**

The paper introduces a new approach to improve the efficiency and robustness of personalized talking face generation. The paper receives unanimous accept from all reviewers. The authors are suggested to further revise according to the comments of the reviewers in the final version.